# Development Control and Inactivation of *Byssochlamys nivea* Ascospores by Hyperbaric Storage at Room Temperature

**DOI:** 10.3390/foods12050978

**Published:** 2023-02-25

**Authors:** Carlos A. Pinto, Diogo Galante, Edelman Espinoza-Suarez, Vítor M. Gaspar, João F. Mano, Francisco J. Barba, Jorge A. Saraiva

**Affiliations:** 1LAQV-REQUIMTE, Department of Chemistry, University of Aveiro, 3810-193 Aveiro, Portugal; 2CICECO—Aveiro Institute of Materials, Department of Chemistry, University of Aveiro, 3810-193 Aveiro, Portugal; 3Nutrition and Food Science Area, Preventive Medicine and Public Health, Food Science, Toxicology and Forensic Medicine Department, Faculty of Pharmacy, Universitat de València, Avda. Vicent Andrés Estellés, 46100 València, Spain

**Keywords:** *Byssochlamys nivea*, mycotoxins, food safety, hyperbaric storage, thermal pasteurization, high pressure processing

## Abstract

This study tested hyperbaric storage (25–150 MPa, for 30 days) at room-temperature (HS/RT, 18–23 °C) in order to control the development of *Byssochlamys nivea* ascospores in apple juice. In order to mimic commercially pasteurized juice contaminated with ascospores, thermal pasteurization (70 and 80 °C for 30 s) and nonthermal high pressure pasteurization (600 MPa for 3 min at 17 °C, HPP) took place, and the juice was afterwards placed under HS/RT conditions. Control samples were also placed in atmospheric pressure (AP) conditions at RT and were refrigerated (4 °C). The results showed that HS/RT, in samples without a pasteurization step and those pasteurized at 70 °C/30 s, was able to inhibit ascospore development, contrarily to samples at AP/RT and refrigeration. HS/RT for samples pasteurized at 80 °C/30 s evidenced ascospore inactivation, especially at 150 MPa, wherein an overall reduction of at least 4.73 log units of ascospores was observed to below detection limits (1.00 Log CFU/mL); meanwhile, for HPP samples, especially at 75 and 150 MPa, an overall reduction of 3 log units (to below quantification limits, 2.00 Log CFU/mL) was observed. Phase-contrast microscopy revealed that the ascospores do not complete the germination process under HS/RT, hence avoiding hyphae formation, which is important for food safety since mycotoxin development occurs only after hyphae formation. These findings suggest that HS/RT is a safe food preservation methodology, as it prevents ascospore development and inactivates them following commercial-like thermal or nonthermal HPP pasteurization, preventing mycotoxin production and enhancing ascospore inactivation.

## 1. Introduction

Molds belong to the kingdom of fungi, which also includes yeasts and common mushrooms. Molds may be classified into six filo: Ascomycota, Basidiomycota, Chytriomycota, Deuteromycota (an informal group of unrelated fungi that solely reproduce asexually), Glomeromycota, and Zygomycota. The Ascomycota produce extraordinarily resistant conidiospores (asexually produced spores) and ascospores (sexually produced spores) [1]. As the ascospores mature, they are released into the air and may travel great distances on the wind, contaminating a broad variety of fields, food-related settings, and eventually food items themselves, posing a hazard to food quality and safety [2]. The extreme resistance of ascospores to thermal and nonthermal processing, oxidative stress, UV-radiation, etc., is due to their exquisite characteristics, such as a protoplast very rich in trehalose and mannitol, which makes their interior very viscous, several heat-shock proteins, and a thick, multi-layered cell wall (>0.5 µm) (an outer and inner cell wall membrane) that protects them from harsh conditions [3,4].

Once the ascospores find favourable environments (nutrients) or the right stimulus (quick heat or pressure-shock), the germination process begins with the hydrolysis of trehalose into glucose monomers, which are released to the external medium, followed by prosilition (separation of the protoplast and the outer cell membrane), water uptake and swelling of the protoplast, metabolism activation, germ tube formation and, consequently, mycelium formation [5,6]. The formation of the mycelium is associated with the production of hazardous secondary metabolites, namely mycotoxins [7]. Mycotoxins have mutagenic, carcinogenic, and genotoxic effects on people and animals; however, they do not cause food poisoning like the cereulide or botulinic toxin of bacteria does. Mycotoxins in food and animal feed are rigorously controlled [8]. The most frequent toxins are patulin (produced by *Penicillium*, *Aspergillus* and *Byssochlamys*), aflatoxins (produced by *Aspergillus*), and ochratoxin A (produced by *Penicillium* and *Aspergillus*), among others [9].

Usually, the most prevalently contaminated foods with ascospores are dairy, fruit juices and their concentrates, jams and grains. These ascospores (especially those from *Aspergillus, Byssochlamys, Penicillium* and *Talaromyces* genera) are able to germinate and develop in a wide range of pH (3–8) and water activities (0.89–0.99) values, and even in environments with low oxygen tensions, causing food spoilage [10]. In this sense, there are some species of fungi, such as those from the species of *Penicillium*, *Talaromyces*, *Aspergillus* and *Byssochlamys* whose ascospores are among the most heat-resistant ones, and can be compared, in particular cases, to a similar level of heat-resistance to that observed for some bacterial spores [11]. 

For instance, the most used strategies to delay ascospore development are either chemical-based hurdles such as pH depletion, low oxygen tensions (vacuum conditions or modified atmosphere packaging), and the addition of chemical preservatives or physical-based hurdles such as low temperatures (refrigeration, freezing, etc), among others [12,13]. Currently, consumers seek minimally processed, clean(er) label food products, which makes the use of chemical-based hurdles to preserve foods difficult, also considering some of the health-issues raised by the use of some chemical preservatives and other food ingredients [14].

Hyperbaric storage (HS), a new food preservation methodology, uses storage under moderate pressure and temperature control for the conventional cold storage-based processes, such as refrigeration. For instance, in a previous work, Pinto and colleagues [15] demonstrated the possibility of inactivating approximately 5 log units of *Alicyclobacillus acidoterrestris* endospores (heat-resistant spore responsible for the spoilage of acidic fruit juices due to the production of off-flavours and odours, with a very important industrial relevance and a huge concern among fruit processors) in apple juice after 48 h at 100 MPa, at RT.

The issues associated with the ascospore-mediated spoilage of foods are very concerning among food processors, considering that they are responsible for considerable economic losses [16] and the potential of HS at RT to inactivate heat resistant microorganisms such as endospores from *A. acidoterrestris*, as previously stated; therefore, it is of interest to evaluate the feasibility of HS to control the germination and development of ascospores. 

Thus, the present study aimed to evaluate HS at uncontrolled RT (18–23 °C), in order to control the germination and development of *Byssochlamys nivea* ascospores inoculated in commercial apple juice (pH 3.7). The storage pressures were set at 25, 50, 75 and 150 MPa, for a maximum evaluation period of 30 days. To simulate a real-case scenario, the inoculated apple juice was pasteurized by different commercial conditions, namely thermal pasteurization (70 and 80 °C for 30 s) and nonthermal HPP (600 Mpa, 3 min, 17 °C), and stored under HS/RT conditions. This way, a real-case scenario of a pasteurized contaminated acidic juice was subjected to HS conditions in order to mimic an industrial problem, looking for HS/RT as a possible solution.

## 2. Materials and Methods

### 2.1. Culture Media and Chemicals

Physiological solution (0.9% NaCl) was acquired from Applichem Panreac (Darmstadt, Germany), and potato dextrose agar (PDA) and potato dextrose broth (PDB) were purchased from VWR (VWR International, Radnor, Pennsylvania, USA).

### 2.2. Ascospore Production, Harvesting and Storage

*B. nivea* ascospores were produced as described by Evelyn and Silva (2018) [17] with minor modifications. Briefly, a single colony from a pure culture of *B. nivea* DSM 1824, purchased from *Deutsche Sammlung von Mikroorganismen und Zellkulturen* (DSMZ, Braunschweig, Germany) was grown in PDB at 30 °C for 96 h at 150 rpm. Afterwards, the liquid culture was sonicated for 5 min in an ultrasonic bath to separate the mould aggregates. Then, 300 µL of the sonicated culture was spread-plated onto PDA plates and incubated at 30 °C for 30 days. The sporulation process was evaluated by phase-contrast microscopy (Zeiss Primovert inverted optical contrast microscope, equipped with a 3MPix monochromatic camera). All images were acquired and post-processed in Zen microscopy software (version 3.0).

After confirming sporulation, the ascospores were harvested by flooding the PDA plates with cold, sterile distilled water, and scratching the surface with a glass rod. Then, the flooded culture was collected and placed in 50 mL falcon tubes filled with 5 mm diameter glass beads. Then, the falcons were vigorously mixed in the vortex for 4–5 min. This procedure allowed the breakage of the asci and release of the entrapped ascospores. Afterwards, the suspension containing both ascospores and the asci was filtered through a sterilized cotton filter to separate the ascospores from the asci. Then, the ascospores were washed 3 times in cold, sterilized distilled water by centrifugation at 5000× *g* for 15 min at 4 °C. The washed ascospores were kept at 4 °C up to 2 weeks to avoid differences in the temperature and pressure resistance that occurs with ascospore aging [6,18].

### 2.3. Ascospore Inoculation and Hyperbaric Storage Conditions

For the present study, the selected inoculation matrix consisted of shelf-stable commercially pasteurized apple juice (pH 3.7, 10 °Brix) that was purchased at a local supermarket and that was aseptically packed (4.7 mL) in low permeability polyamide–polyethylene bags with a thickness of 90 μm (PA/PE-90, IdeiaPack–Comércio de Embalagens, Lda, Viseu, Portugal); these were previously sterilized by UV-radiation for 15 min in a laminar flow chamber (BioSafety Cabinet Telstar Bio II Advance, Terrassa, Spain) to avoid contaminations. Afterwards, 300 μL of *B. nivea* ascospores were added to the juice under aseptic conditions, reaching a final concentration of approximately 5 Log CFU/mL of ascospores.

### 2.4. Pre-Activation

In order to determine the influence of pre-activation strategies (commercial-like thermal and nonthermal pasteurization procedures), thermal pasteurization (70 and 80 °C for 30 s, performed in a water bath) and nonthermal HPP (600 MPa, 3 min, 17 °C, performed in an industrial-scale HPP unit (Hiperbaric 55L, Hiperbaric SA, Burgos, Spain)) were performed. This procedure aimed to evaluate how the ascospores, in a previously pasteurized fruit juice, would behave afterwards under HS conditions. At the same time, inoculated samples without any pre-activation step were also placed under HS conditions.

### 2.5. Hyperbaric Storage Conditions

The inoculated samples (with and without a pre-activation step) were placed in HS equipment (Stansted Fluid Power FPG13900, Stansted, UK) equipped with a pressure vessel that was 30 mm in inner diameter and 250 mm in height, using a mixture of propylene glycol and water (40:60 *v*/*v*) as pressurization fluid. The storage pressures were set at 25, 50, 75 and 150 MPa at naturally variable/uncontrolled RT (18–23 °C). At the same time, control samples were kept at atmospheric pressure (0.1 MPa, AP), at RT and under refrigeration (4 °C), immersed in the same pressurization fluid and kept in the dark.

### 2.6. Determination of Ascospore Germination and Inactivation

After each storage condition, 1.0 mL juice samples were serially diluted in 0.9% physiological solution, which were afterwards spread onto PDA [19]. At the same time, an aliquot of juice was heat treated at 70 °C for 10 min to inactivate the vegetative forms [20,21], and then also serially diluted in the aforementioned solution and plated on PDA plates. This procedure allowed to determine the thermal resistant fraction, i.e., ungerminated ascospores. Considering that, in the first case, both vegetative forms and ascospores could have been present, this fraction will be termed the total microbial load, while the heat-treated fraction will be termed the thermal resistant fraction (that contains only ascospores). The colonies in the plates were considered countable when the number of colonies ranged between 10-100. The quantification limit of 2.00 Log CFU/mL (corresponding to a number of colonies between 1 and 9) and the detection limit of 1.00 Log CFU/mL (when no colonies were found on the plates) were established.

### 2.7. First-Order Kinetic Model of B. nivea Ascospores Inactivation

The decimal reduction time (D_t_-values, time required at a certain pressure to reduce the microbial population by 90%) was determined from the reciprocal of the slope [22], for the cases wherein ascospore inactivation was observed with a sufficient number of experimental points (minimum of three), according to Equation (1):(1)Log NN0=⁡-tDt
wherein N_0_ and N represent the initial spore population in each matrix and the number of survivors after being exposed under lethal HS conditions for a certain period of time. 

The pressure coefficient (z_p_, expressed in MPa and representing the pressure increase that results in a 10-fold increase in the D-value) was determined according to Equation (2):(2)Log DDref=⁡-pref-pzp
wherein D_ref_ represents the D-value at the reference pressure, while p_ref_ and p represent the reference pressure and HS-set pressure.

The data fitting was performed using Matlab R2021a software (MathWorks Inc., Natick, MA, USA). The results are displayed as Appendix A, in Appendix A. To evaluate the quality of each model, the mean square root error (*MSRE*) and coefficient of determination (*R^2^*) were determined. A *MSRE* close to 0 and a *R^2^* close to 1 indicate the adequacy of the models to describe the data.

### 2.8. Phase-Contrast Microscopy

In order to observe ungerminated and germinated/dead ascospores, phase-contrast microscopy was used, with the first group appearing as bright phase ascospores in a dark field due to the relatively high cell density; meanwhile, the second group appears as a dark phase ascospore due to water uptake and the loss of solutes, which lower the spore density [11]. The pictures were taken at the 30th day of each storage condition and compared with the initial samples, using a Zeiss Primovert inverted optical contrast microscope, equipped with a 3MPix monochromatic camera. All images were acquired and post-processed in Zen microscopy software.). 

### 2.9. Statistical Analyses

The microbiological analyses were performed in duplicate, each one from triplicated samples. The obtained results are expressed as mean ± standard deviation and were statistically analyzed using a one-way Analysis of Variance (ANOVA), followed by Tukey’s Honestly Significant Difference (HSD) test at a significance level of 5%.

## 3. Results

### 3.1. B. nivea Ascospores without a Pre-Activation Step

At the beginning of the storage experiments with samples with no pre-activated ascospores, the initial total microbial load was 5.23 ± 0.01 Log CFU/mL, while the thermal resistant fraction presented a total load of 4.17 ± 0.02 Log CFU/mL (Figure 1). This difference may be attributed to some vestigial vegetative cells that could have remained after the harvesting process.

At AP/RT, hyphae formation was evident (even to the naked eye) after 30 days of the storage experiments, despite only a slight increase (*p* < 0.05) in the total microbial load (of about 0.32 log units) being observed. Meanwhile, for samples kept under refrigeration, for the same period, an even smaller increase (*p* < 0.05) of 0.19 log units was observed; yet, hyphae formation was only visible under phase-contrast microscopy (Figure 2), as will be discussed further. A less pronounced and slow increase in the *B. nivea* count was observed in the aforesaid conditions because the ascospores were not activated, i.e., the germination process could only be activated by nutrients; this is normally a slower process when compared to physically induced ascospore germination [6], or does not even occur, as some ascospores can only be activated by very high temperatures and/or hydrostatic pressures [23]. 

A different scenario was observed for samples kept under HS/RT. Indeed, a general reduction in the total microbial load of about 0.58, 1.25 and 1.42 log units was observed at 25, 50 and 75 MPa, respectively, after 30 days of storage, as seen in Figure 1A. A similar scenario was found for the thermal resistant fraction, despite some local variations depending on the storage pressure; yet, the decrements were lower (Figure 1B) compared to those observed for the total microbial load.

Interestingly, at 150 MPa, there was a significant decrease (*p* < 0.05) in the total microbial load of about 1.14 log units by the 5th day of the storage experiments; nevertheless, an increase in the total microbial load and thermal resistant fraction was observed (of about 0.5 and 0.6 log units, respectively) from the 5th day onwards. This increase in the total ascospore load is most likely due to the release of entrapped ascospores that could have remained inside the asci as a consequence of the harvesting process [5,24], as it is also accompanied by an increase in the thermal resistant fraction (Figure 1B).

Nonetheless, no hyphae formation was observed, even under phase-contrast microscopy, after 30 days at 150 MPa (Figure 2), showing the clear effect of storage pressure on limiting hyphae formation. 

The following sections will display the results obtained with previously pasteurized apple juice containing ascospores, which was pasteurized using commercial conditions currently applied in industry and that can result in vegetative microorganisms’ inactivation, leaving the ascospores in case they are present.

### 3.2. B. nivea Ascospores Pre-Activated by Thermal Pasteurization

With regard to samples that had been pre-activated by thermal pasteurization, it is noteworthy that a different evolution pattern was evident depending on the storage condition and on the temperature the pre-activation process took place at. 

As a consequence of the heat activation at 70 °C for 30 s, a decrease of about 0.98 log units was observed in the total microbial load, possibly due to the inactivation of the vegetative cells that remained after the harvesting process of the ascospores [21]. Thus, the initial total microbial load and total thermal resistant fraction load values at the beginning of the storage experiments were 4.22 ± 0.04 and 4.33 ± 0.08 Log CFU/mL, respectively, as displayed in Figure 3A,B.

Immediately after 5 days under AP/RT conditions, an increase (*p* < 0.05) in the total microbial load of about 1.08 log units (Figure 3A) was observed, which was accompanied by a decrease (*p* < 0.05) in the thermal resistance (Figure 3B), due to the germination and development of the ascospores. In fact, hyphae formation was noticeable in these samples, even to the naked eye. For refrigerated samples, after 10 days of storage, an increase (*p* < 0.05) of about 0.77 log units was noticed, which was also accompanied by a decrease (*p* < 0.05) in the thermal resistance fraction (Figure 3B). 

When it comes to HS/RT samples, globally, and despite some punctual variations among the storage conditions, there was an increase (*p* < 0.05) in the total microbial load, which was generally accompanied by an increase (*p* < 0.05) in the thermal resistant fraction. This could be related to the release of entrapped ascospores that could have remained trapped on the inside of the asci during the harvesting process and could have been disrupted by hydrostatic pressure. In addition, despite an increase in both the total ascospore load and the thermal resistant fractions, no hyphae formation was observed for samples stored under HS/RT conditions, regardless of the storage pressure. This observation is particularly relevant for controlling ascospore development and, ultimately, hyphae formation, which can lead to the production of mycotoxins [9,25].

Pre-activation by thermal processing at 80 °C for 30 s resulted in distinct evolution patterns for the different storage conditions. The thermal pre-activation step resulted in a small increase (*p* < 0.05) in the total microbial load (about 0.3 log units, Figure 4A), while the total ascospore load remained unchanged (*p* > 0.05) (Figure 4B).

At AP/RT, an increase (*p* < 0.05) in the total ascospore load was observed, which was accompanied by a reduction (*p* < 0.05) in the thermal resistant fraction (from the 1st to the 2nd day of the storage experiments); this can be associated with ascospore germination and development, leading ultimately to hyphae formation (Figure 5). For this storage condition, the microbiological analyses were performed only until the 10th day, due to severe hyphae formation, and with the total microbial load reaching 6.41 Log CFU/mL. The recovery of the thermal resistant fraction from the 2nd day onwards may be related to the formation of new ascospores that were trapped within the asci of the hyphae (Figure 5A), which could have been released and activated as a consequence of the thermal processing that was performed in order to quantify the thermal resistant fraction.

Regarding the refrigerated storage, a gradual increase in the total microbial load was observed depending on the storage, which was accompanied by an increase in the thermal resistant fraction, at least until the 20th day of the storage experiments. At the 30th day, a stabilization of the total microbial load (reaching a maximum load of 5.78 ± 0.10 Log CFU/mL) was observed, which was accompanied by a decrease (*p* < 0.05) in the thermal resistant fraction; this storage methodology inhibited, temporarily, ascospore germination and development. The formation of very small hyphae was noticed under phase-contrast microscopy (Figure 5B).

Considering HS/RT, there was an overall reduction in the total microbial load from the first day of storage onwards (except at 25 MPa, for which the total microbial load was similar (*p* > 0.05) to the initial load by the 30th day). At 25 and 50 MPa, an increase in the thermal resistant fraction was also observed, possibly due to the gradual release of entrapped ascospores from the asci. Yet, at 50 MPa, an overall reduction in the total microbial load, of ≈1.11 log units, was observed by the 30th day of storage. A more interesting scenario was observed at 75 and 150 MPa, wherein a gradual reduction (*p* < 0.05) in both the total microbial load and the thermal resistant fraction were observed; the quantification limit (2.00 Log CFU/mL) was reached after 15 days of storage (for both pressures), and at 150 MPa, the detection limit (1.00 Log CFU/mL) was even reached by the 30th day of storage. This represents a reduction of at least 3.77 log units (considering the initial load at the beginning of the storage experiments) or 4.73 log units, considering the maximum total microbial load observed at the first day of storage, wherein the release of entrapped ascospores may have occurred. Generally, at 75 and 150 MPa, and for the same storage days for each pressure, the total microbial load was higher than the total resistant fraction; this suggests that hydrostatic pressure not only limited ascospore development, but also targeted these structures, by means of sensitizing them to the subsequent heat-shock that was performed to inactivate the germinated ascospores (ascospores with no thermal resistance). Additionally, the inactivation curves observed at 75 and 150 MPa resemble the non-linear Lorentzian model used to describe the inactivation pattern of several heat-resistant ascospores that are activated during the first stages of thermal and nonthermal processing (initial increase on the microbial load), followed by a decrease over time [26,27]. Due to the reduced number of quantifiable microbiological values, an accurate estimation of the aforementioned model parameters was not possible.

### 3.3. B. nivea Ascospores Pre-Activation by HPP

Immediately after pasteurization at 600 MPa for 3 min, an initial decay in the total microbial load of about 0.88 log units was observed, possibly due to the presence of vestigial vegetative cells that remained after the washing process and that were destroyed by HPP. Therefore, the total microbial load at the beginning of the storage experiments was 4.29 ± 0.13 Log CFU/mL, as seen in Figure 6A. The thermal resistant fraction was also reduced from 4.24 ± 0.11 to 2.67 ± 0.06 Log CFU/mL, suggesting that the initial HPP treatment considerably reduced the thermal resistance of the ascospores (Figure 6B).

Samples stored at AP/RT conditions quickly underwent ascospore germination and development, with evident hyphae formation by the 10th day of the storage experiments, where a maximum load of 5.56 ± 0.02 Log CFU/mL was reached. For this condition, the thermal resistant fraction load also increased (*p* < 0.05) depending on the storage, due to the formation of new ascospores because of hyphae formation, which were visible to the naked eye. Regarding the refrigerated samples, the total microbial load was gradually reduced depending on the storage, yet a recovery was observed by the 30th day of the storage experiments, which was accompanied by a small increase (*p* < 0.05) in the thermal resistant fraction. Even though no hyphae formation could be observed by the naked eye, the formation of small hyphae was evident when the samples were observed under the phase-contrast microscope (Figure 7).

Contrarily, samples stored under HS/RT faced a quick reduction in the total microbial load, which seems to be proportional to the storage pressure level, i.e., at higher pressures, both the total microbial load and thermal resistant fraction were more prone to inactivation. Indeed, after 5 days of storage at 75 and 150 MPa, the quantification limit (2.00 Log CFU/mL) was reached, representing a reduction of about 2.3 log units, achieving a similar level of inactivation at 50 MPa after 15 days of the storage experiments. At 25 MPa in the first 5 days of storage, a gradual and steady decay in the total microbial load was observed (a reduction of about 1.8 log units), and, on the remaining days, the total microbial load remained unchanged (*p* > 0.05) until the end of the storage experiments. Yet, no consistent pattern was observed for the thermal resistant fraction, which was ultimately reduced below the quantification limit after 20 days of storage, remaining as such until the end of the storage experiments. In addition, no hyphae formation was observed for any of the samples stored under HS/RT even under phase-contrast microscopy (Figure 7 displays the results for samples stored at 150 MPa for 30 days); this is in contrast to AP/RT and refrigerated storage, as aforementioned. At 50 MPa, a biphasic inactivation was observed, with an accelerated microbial load decay in the first two days, and then a slower decay from the second day onwards (until the 15th day of storage, wherein the quantification limit was reached).

When compared to thermal activation at 80 °C/30 s, HPP at 600 MPa for 3 min resulted in a more extensive ascospore sensibilization, as the lower pressure, 50 MPa, was able to gradually reduce the ascospore counts depending on the storage, with the absence of the typical “shoulders” displayed on the ascospore inactivation curves. In addition, the D-values (available as Appendix A) observed at 75 MPa for the HPP-activated ascospores were lower compared to the ones observed for the thermally activated ascospores (0.985 ± 0.081 vs. 2.502 ± 0.711 days, respectively); meanwhile, at 150 MPa, the D-values were practically the same for the HPP and the thermally activated ascospores (1.030 ± 0.564 vs. 1.112 ± 0.040, respectively), as displayed in Appendix A (available as Appendix A). It is also noteworthy that the quantification limit was reached after 5 days (at 75 and 150 MPa) for the samples previously pasteurized by HPP, while the same limit was reached after 15 days (also at 75 and 150 MPa) for the samples thermally pasteurized at 80 °C/30 s. Notwithstanding, the detection limit (1.00 Log CFU/mL) was only reached for samples previously pasteurized at 80 °C/30 s and kept at 150 MPa for 30 days, representing an ascospore inactivation of 4.72 log units (considered from the 1st day of storage onwards).

### 3.4. Phase-Contrast Microscopy

The general aspect of the ascospores before and after each pre-activation (pasteurization) technique is displayed in Appendix A (available as Appendix A). As the microbiology results obtained for the activation condition of 70 °C/30 s were similar to the non-activated ascospores, phase-contrast microscopy images were not taken for the first condition. It is noteworthy that bright-phase analysis revealed an increase in the number of ascospores after thermal activation at 80 °C/30 s (Appendix A), when compared to the other pre-activation procedures. Very vestigial vegetative cells were also noticeable, which correlates with the cases in which some initial inactivation of the total microbial load was observed. Samples not pre-activated and samples pre-activated at 70 °C/30 s (Appendix A) seemed to result in mild ascospore aggregation (more evident for the latter condition), while conditions of 80 °C/30 s and 600 MPa/3 min seemed to exert a dispersive effect. This phenomenon is known to be caused by the electrostatic attractivity [28], which can be broken by physical methods, such as high temperatures and HPP [24].

The formation of hyphae is noteworthy, as these were observable even to the naked eye on samples stored at AP/RT conditions, regardless of the pre-activation process; meanwhile, for samples kept under refrigeration, even though no hyphae formation was observed by the naked eye, their formation was evident by phase-contrast microscopy, showing that ascospores can develop under refrigeration conditions, and, even worse, produce mycotoxins such as patulin, byssochlamic acid and byssotoxin A [29].

Contrarily, samples stored under HS/RT, especially at 150 MPa, were revealed to limit hyphae formation and cause ascospore inactivation, as highlighted in Figure 2C, Figure 5C and Figure 7C, regardless of the pre-activation strategy. This is of particular interest considering that toxin formation can be avoided by avoiding ascospore germination and development. For the cases in which ascospore inactivation occurred, the presence of dark phase ascospores is noticeable, especially in Figure 5C and Figure 7C, evidencing the existence of dead ascospores in the samples.

Another point relates to the presence of bright-phase ascospores in heat-activated samples (80 °C for 30 s) after 30 days at 150 MPa, even though both the total microbial load and total thermal resistant fraction were below the detection limits (1.00 Log CFU/mL); these could afterwards germinate and develop in case the pressure hurdle was eliminated. These ascospores, even though they were not cultivable, could slowly germinate and develop. For example, Bermejo-Prada and colleagues [30] reported the recovery of yeasts and molds in strawberry juice after being conditioned for 3 days at AP and 20 °C, considering that these samples were previously under HS conditions (50 MPa, 15 days, 20 °C) and that the microbial counts were below detection limits. Another study from Fidalgo and colleagues [31] also reported the recovery of yeasts and molds in watermelon juice (to values above 6 Log CFU/mL) previously placed under HS conditions (100 MPa, 60 h, RT) (and whose yeasts and molds counts were below detection limits) after being placed under refrigeration (5 °C) conditions for 12 days. The aforementioned recovery processes could be hypothesized to be due to the presence of a small population of ascospores on both juices, which were not inactivated during HS; however, the authors did not provide such information.

## 4. Discussion

### 4.1. B. nivea Ascospores without a Pre-Activation Step

It is noteworthy that the effects of low/moderate hydrostatic pressures on fungi spores are scarcely reported in the literature; however, Eicher and Ludwig [32] studied the effects of low pressure exposure (200 MPa for 1h at 25 °C) on *Eurotium repens* ascospores inoculated in an isotonic NaCl solution (fortified with 0.1% Tween 80), and observed only a small ascospore activation (small increase in the ascospore load). In another study, Maggi and colleagues [33] reported a small decrease in *B. nivea* ascospores that were inoculated in physiological solution (8.5 g/L NaCl) loads after a moderate-pressure treatment at 300 MPa for 1 h at 25 °C; the author attributed this to the inactivation of vestigial vegetative cells.

Based on previous studies with HS/RT, it is generally reported that the class of yeasts and molds is gradually considerably inactivated depending on the storage [34,35]. Considering an example of an acidic product, strawberry juice (pH 3.3), Bermejo-Prada and colleagues [30] reported a decrease of about 2 log units in yeasts and molds after 10 days at 50 MPa and a decrease of 0.8 log units after 6 days at 100 MPa, which are considerably higher inactivation levels than those observed in the present work (Figure 1A). Notwithstanding, a direct comparison cannot be fully established, considering that the aforementioned study regarded indigenous yeasts and molds of strawberry juice (pH 3.3 vs. pH 3.7 in the present study). Yet, this statement intends to highlight that higher inactivation levels were expected, especially at higher pressures.

A closer look at the thermal resistant fraction (Figure 1B) allows the reader to encounter no major variations in the overall thermal resistance of the ascospores, regardless of the storage condition (either at AP or under HS conditions). The pressure resistance of the non-pre-activated ascospores is due to the high amounts of trehalose within the protoplast, which comprises between 9 to 17% of the ascospore dry weight and which makes the protoplast extremely viscous [36]; this is also due to a thick multi-layered cell wall that protects the ascospore from physical and chemical damages, and which does not allow the proper transmission of pressure.

The range of evaluated pressures (25–150 MPa) is not suitable to break spore dormancy and trigger germination under these pressure; this is similar to the results observed for other spores, such as those from bacteria. This includes, for instance, those from *Bacillus* spp., which usually undergo nutrient-like physiological germination up to 300–400 MPa. This phenomenon is caused by the triggering of the germinant receptors caused by hydrostatic pressure, resulting in the initiation of germination and a loss of thermal resistance and outgrowth, considering that it is a rather quick mechanism (a few minutes or hours, depending on the pressure level and temperature) [37]. Therefore, given the fact that the thermal resistant fraction of the ascospores is poorly affected, low pressures up to 150 MPa are less likely to trigger *B. nivea* ascospore germination; this is also visible on the phase-contrast microscopy images of the spores after 30 days under HS conditions. 

Yet, more fundamental research is needed to fully understand the possibility of triggering the germination process of ascospores using mild hydrostatic pressures, even though this study could be a first insight into this issue. Even though no ascospore inactivation occurred, it was possible to avoid hyphae formation; this is of extreme importance for food safety, as this avoids mycotoxin formation, and highlights the possibility of using HS at uncontrolled RT (thus practically without energetic costs) to store, for example, raw products.

### 4.2. B. nivea Ascospores with a Thermal and Nonthermal Pasteurization Step

For samples pre-activated at 80 °C/30 s, as abovementioned, there was an increase (*p* < 0.05) in the total microbial load (Figure 4A) after a thermal pasteurization at 80 °C for 30 s. A similar observation was made by Evelyn and Silva (2015) [38], who processed strawberry puree (pH 3.4, 8.1 °Brix) at 75 °C for 30 min, and reported an increase of 1.2 log units on the ascospore counts, and termed this increase as “shoulders”. In another study, Ferreira and colleagues [39] also reported an increase in the ascospore loads (*B. nivea* and *B. fulva*) of about 0.5 log units after thermal processing at 85 °C for 10 min. Both authors attributed this behavior to the effects of heat as a mechanism to activate dormant ascospores (that otherwise would remain dormant or present a very slow germination) if triggered by nutrients or other chemicals [40,41]. 

At 75 and 150 MPa, the total microbial load was reduced below quantification limits after 15 days of storage for both pressures, which represents an overall reduction of at least 4 log units. This is remarkable considering that only a short heat-treatment at 80 °C/30 s was performed, followed by HS at RT, and yet this resulted in ascospore inactivation depending on the storage, without the use of very intense thermal processing. This means that HS can not only be used as a safe storage methodology, by avoiding ascospore development, but also for ascospore inactivation. To put this into context, the same level of inactivation (≈4 log units) was only achievable by very intense thermal processing at 90 °C for 6–8 min and for more than 45 min at 85 °C in strawberry puree [38], or by combining high hydrostatic pressure and high temperature (600 MPa, 15 min, 80 and 90 °C) in apple juice and puree, respectively [42]. Another study with *Byssochlamys nivea* ascospores reported that these ascospores survived a very intense thermal process of 103 °C for 7 min in pineapple nectar [39], or in tomato juice processed at 90 °C for 20 min [43]. This can have detrimental consequences for the food product, or, from an application point of view, be harsh for heat sensible products (either food or, for example, cosmetic or pharmaceutical products).

Regarding HPP-pre-activated samples, a reduction in the thermal resistant fraction was observed immediately after the HPP process; this can be related to structural damages on the crust, according to the literature, which will ultimately facilitate the ascospore hydration and an efficient pressure-transmission afterwards [6]. In addition, it is remarkable that, after 5 days of storage at 75 and 150 MPa, the total microbial load and total thermal resistant fraction were below quantification limits, representing a reduction of approximately 2.5 log units after 5 days. These results highlight the fact that it is possible to inactivate *B. nivea* ascospores without using high temperatures; this is of importance for either heat sensible foods, or can have implications for other value-added products, such as cosmetic or pharmaceutical products [24].

To put these results into perspective, the combination of high pressures and moderate/high temperatures may not be enough to fully inactivate *B. nivea* ascospores, as observed by Evelyn and colleagues, who combined high pressures and temperatures (600 MPa, 10 min, 75 °C) in ascospores with different ages (ranging between 4 and 12-week-old ascospores) and observed inactivation levels ranging from 2.5 and 3.5 log units, depending on the ascospores’ age [17]. In another study, Ferreira and colleagues reported viable ascospores in pineapple nectar after a combined treatment of high pressure and temperature (600 MPa for 10 min at 90 °C) [42]. Considering these results, it can be stated that a similar level of inactivation was achieved in the present study by very short thermal exposure, or even without the application of heat.

Overall, HS/RT could be seen as a post-pasteurization tool to inactivate the heat-resistant ascospores that survive the conventional thermal [44] and nonthermal pasteurization procedures; then, the food product could be placed at room temperature, as it is currently performed for pasteurized acidic products such as apple juice. In this way, the elimination of ascospores can be ensured, resulting in higher food safety, as hazards, such as the formation of mycotoxins formation, would be avoided. 

The findings of the present work should yet be deepened and confirmed at an industrial-scale level to infer the true potential of HS at uncontrolled RT to inactivate *B. nivea* ascospores (and even other ascospores), namely after removing samples from HS conditions and performing a shelf-life evaluation.

## 5. Conclusions

In this study, the feasibility of employing HS/RT to control ascospore germination and development was demonstrated, and the influence of both thermal and nonthermal pasteurization processes on subsequent ascospore control was assessed. In contrast to conventional refrigeration, the findings indicate that HS/RT effectively prevented the germination and growth of ascospores by preventing the creation of hyphae. A pre-activation step (in the range of pasteurization conditions) at 70 °C resulted in an overall evolution of the total microbial load (and the thermal resistant fraction) that was comparable to that observed for unpasteurized samples (non- pre-activated ascospores); meanwhile, a pasteurization at 80 °C for 30 s, and at 600 MPa for 3 min (17 °C) by HPP, followed by HS/RT conditions, resulted in not only in ascospore development control, but also in ascospore inactivation, especially at higher HS pressures (75 and 150 MPa). These findings are relevant in light of the potential of inactivating heat-resistant ascospores of *B. nivea* at room temperature, using commercial-style thermal pasteurization, or even without applying heat (by non-thermal HPP pasteurization). In addition, the coupling of a preceding nonthermal HPP pasteurization stage with HS (e.g., 75 or 150 MPa) is a potential approach for inactivating heat-resistant ascospores in thermally labile food items or other thermolabile goods, such as pharmaceutical or cosmetic products.

## Figures and Tables

**Figure 1 foods-12-00978-f001:**
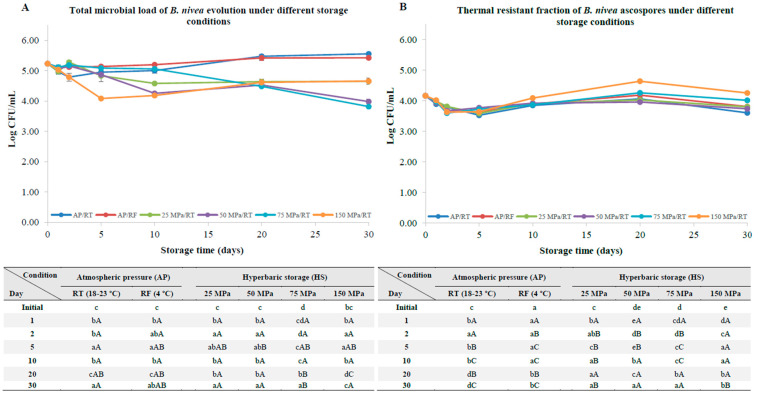
Total microbial load (unheated samples, (**A**)) and thermal resistant ascospore fraction (heat-treated samples at 70 °C for 10 min, (**B**)) of *B. nivea* in commercial apple juice (pH 3.7) stored at atmospheric pressure (AP) and naturally variable/uncontrolled room temperature (18–23 °C, AP/RT), AP under refrigeration (4 °C, AP/RF) and hyperbaric storage (25, 50, 75 and 150 MPa, HS) at naturally variable/uncontrolled RT, without any pre-activation step. Different lower-case letters (a–e) indicate significant differences (*p* < 0.05) between storage days, while different upper-case letters (A–C) indicate significant differences between storage conditions (for a given day).

**Figure 2 foods-12-00978-f002:**
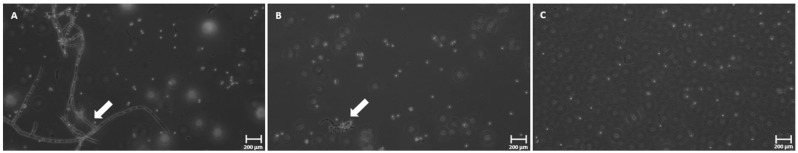
Phase-contrast microscopy images (20× magnification) of the commercial apple juice inoculated with *B. nivea* ascospores in unpasteurized samples stored for 30 days at atmospheric pressure (AP) at room temperature (RT) (**A**), under refrigeration at 4 °C (**B**) and under hyperbaric storage at 150 MPa at RT (**C**). The white arrow highlights hyphae formation.

**Figure 3 foods-12-00978-f003:**
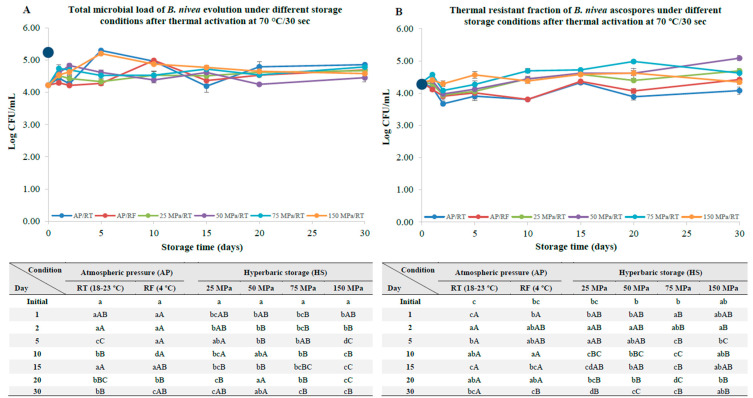
Total microbial load (unheated samples, (**A**)) and thermal resistant ascospore fraction (heat-treated samples at 70 °C for 10 min, (**B**)) of *B. nivea* in commercial apple juice (pH 3.7) stored at atmospheric pressure (AP) and naturally variable/uncontrolled room temperature (18–23 °C, AP/RT), AP and refrigeration (4 °C, AP/RF) and hyperbaric storage (25, 50, 75 and 150 MPa, HS) at naturally variable/uncontrolled RT, without a thermal activation step at 70 °C for 30 s. Different lower-case letters (a–c) indicate significant differences (*p* < 0.05) between storage days, while different upper-case letters (A–C) indicate significant differences between storage conditions (for a given day). The dark and larger circle at the Y axis represents the initial load before the thermal activation step at 70 °C/30 s. The dark blue circle at the Y axis represents the initial load before the thermal activation step at 70 °C/30 s.

**Figure 4 foods-12-00978-f004:**
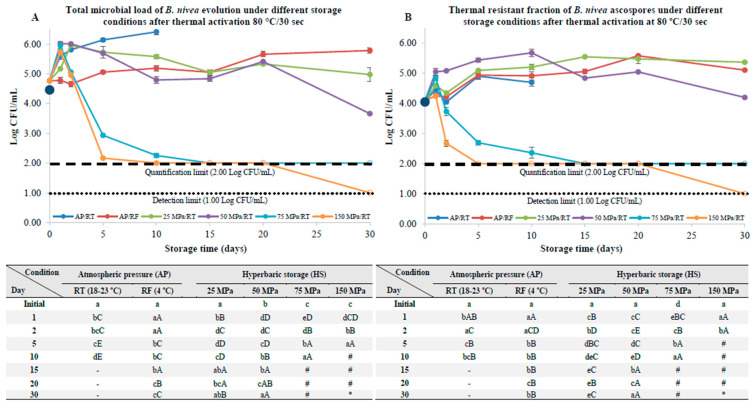
Total microbial load (unheated samples, (**A**)) and thermal resistant ascospore fraction (heat-treated samples at 70 °C for 10 min, (**B**)) of *B. nivea* in commercial apple juice (pH 3.7), stored at atmospheric pressure (AP) and naturally variable/uncontrolled room temperature (18–23 °C, AP/RT), AP and refrigeration (4 °C, AP/RF) and hyperbaric storage (25, 50, 75 and 150 MPa, HS) at naturally variable/uncontrolled RT, without a thermal activation step at 80 °C for 30 s. Different lower-case letters (a–e) indicate significant differences (*p* < 0.05) between storage days, while different upper-case letters (A–E) indicate significant differences between storage conditions (for a given day). Cases marked with “-” indicate samples not analyzed due to severe spoilage, and with “#” and with “*” refer to samples that reached the quantification limit (of 2.00 Log CFU/mL) and the detection limit (1.00 Log CFU/mL), respectively. The dark and larger circle at the Y axis represents the initial load before the thermal pre-activation step at 80 °C/30 s. The dashed and dotted lines indicate the quantification and detection limits that were reached, respectively.

**Figure 5 foods-12-00978-f005:**
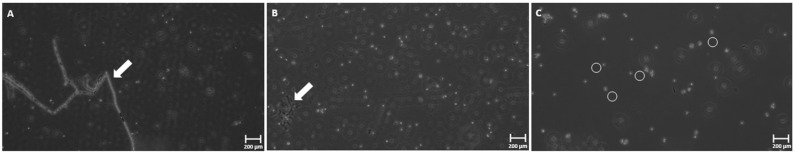
Phase-contrast microscopy images (20× magnification) of the commercial apple juice inoculated with *B. nivea* ascospores in spores activated at 80 °C/30 s stored for 10 days at atmospheric pressure (AP) at room temperature (RT) (**A**), and 30 days under refrigeration at 4 °C (**B**) and under hyperbaric storage at 150 MPa at RT (**C**). The white arrow highlights hyphae formation, while the white circles highlight dead ascospores (displayed as dark phase ascospores).

**Figure 6 foods-12-00978-f006:**
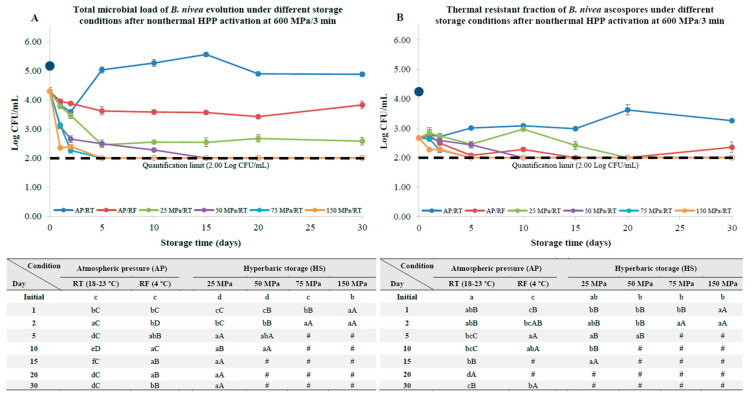
Total microbial load (unheated samples, (**A**)) and thermal resistant ascospore fraction (heat-treated samples at 70 °C for 10 min, (**B**)) of *B. nivea* in commercial apple juice (pH 3.7) stored at atmospheric pressure (AP) and naturally variable/uncontrolled room temperature (18–23 °C, AP/RT), AP and refrigeration (4 °C, AP/RF) and hyperbaric storage (25, 50, 75 and 150 MPa, HS) at naturally variable/uncontrolled RT, with a nonthermal activation-step at 600 MPa for 3 min at 17 °C. Statistical analysis available as Appendix A. Different lower-case letters (a–f) indicate significant differences (*p* < 0.05) between storage days, while different upper-case letters (A–D) indicate significant differences between storage conditions (for a given day). Cases marked with “#” mean that the quantification limit (of 2.00 Log CFU/mL) was reached, and cases marked with “*” mean that the detection limit (1.00 Log CFU/mL) was reached. The dark blue and larger circle at the Y axis represents the initial load before the nonthermal HPP activation step at 600 MPa/3 min. The dashed line means that the quantification limit was reached.

**Figure 7 foods-12-00978-f007:**
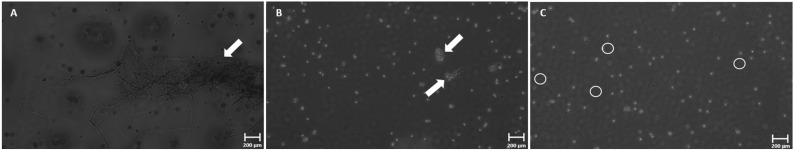
Phase-contrast microscopy images (20X magnification) of the commercial apple juice inoculated with *B. nivea* ascospores in spores activated by HPP at 600 MPa for 3 min at 17 °C, stored for 30 days at atmospheric pressure (AP) at room temperature (RT) (**A**), under refrigeration at 4 °C (**B**) and under hyperbaric storage at 150 MPa at RT (**C**). The white arrow highlights hyphae formation, while the white circles highlight dead ascospores (displayed as dark phase ascospores).

## Data Availability

The data from the present manuscript are available upon request to the corresponding author.

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
