# Peer review of "Development Control and Inactivation of Byssochlamys nivea Ascospores by Hyperbaric Storage at Room Temperature"

_foods, 2023, doi:10.3390/foods12050978_

Round 1

Reviewer 1 Report

Reviewer comments

Manuscript title: Byssochlamys nivea ascospores development control and inactivation by hyperbaric storage at room temperature

This study aimed to evaluate  hyperbaric storage at uncontrolled room temperature to prevent the germination and development of Byssochlamys nivea ascospores inoculated in commercial apple juice (pH 3.7) in comparison with the standard food preservation methods.

The results of the manuscript is interesting. However, the manuscript needs minor revision.

1-English usage needs revision and some parts are difficult to understand.

As examples

Line 59 (hydrolysis of trehalose in two glucose monomers) correct to

(hydrolysis of trehalose into glucose monomers)

Line 60 (Media) correct to (medium)

Line 107 (the potential of HS at RT to inactivate at RT heat) correct to (the potential of HS at RT to inactivate heat)

Line 108 (as such as) correct to (such as)

Line 130 (was spread-plated onto PDA plates, and then incubated) correct to (was spread onto PDA plates, and incubated)

Line 142 (3-fold centrifugation) (what is mean??). Do you mean 3 times centrifugation? Correct please

Line 152 ( in aseptic) correct to (under aseptic)

Line 154 (Pre-activation methodologies) correct to (Pre-activation)

Line 172 (spread plated) correct to (spread)

Line 174 (plated in) correct to (plated on)

Line 179 ( ascospores The) correct to (ascospores. The)

Line 268 (In what regards ) correct to ( with regard to)

Line 345 delete (In spite of not being seen by naked eye,)

Line 368 delete (Although)

General comments

·        pH 3.30 as example  correct to pH 3.3

               please correct pH values through the manuscript

·        Maggi and colleagues (1994) [39]

I think the year will be deleted and the correct is

               Maggi and colleagues [39]

              Please modify their similarities through the manuscript

Sincerely

Author Response

#Referee 1

This study aimed to evaluate  hyperbaric storage at uncontrolled room temperature to prevent the germination and development of Byssochlamys nivea ascospores inoculated in commercial apple juice (pH 3.7) in comparison with the standard food preservation methods.

The results of the manuscript is interesting. However, the manuscript needs minor revision.

English usage needs revision and some parts are difficult to understand. As examples Line 59 (hydrolysis of trehalose in two glucose monomers) correct to (hydrolysis of trehalose into glucose monomers)

Changed accordingly.

Line 60 (Media) correct to (medium)

Changed accordingly.

Line 107 (the potential of HS at RT to inactivate at RT heat) correct to (the potential of HS at RT to inactivate heat)

Changed accordingly.

Line 108 (as such as) correct to (such as)

Changed accordingly.

Line 130 (was spread-plated onto PDA plates, and then incubated) correct to (was spread onto PDA plates, and incubated)

Changed accordingly.

Line 142 (3-fold centrifugation) (what is mean??). Do you mean 3 times centrifugation? Correct please

Changed accordingly. Yes, we intended to say that the centrifugation process was performed three times. This information was clarified in the manuscript. Now it reads, in lines 140-141: “…the ascospores were washed 3 times in cold, sterilized distilled…”.

Line 152 (in aseptic) correct to (under aseptic)

Changed accordingly.

Line 154 (Pre-activation methodologies) correct to (Pre-activation)

Changed accordingly.

Line 172 (spread plated) correct to (spread)

Changed accordingly.

Line 174 (plated in) correct to (plated on)

Changed accordingly.

Line 179 (ascospores The) correct to (ascospores. The)

Changed accordingly.

Line 268 (In what regards) correct to (with regard to)

Changed accordingly.

Line 345 delete (In spite of not being seen by naked eye,)

Deleted as suggested.

Line 368 delete (Although)

Deleted as suggested.

General comments

  • pH 3.30 as example  correct to pH 3.3: please correct pH values through the manuscript

Corrected as suggested.

  • Maggi and colleagues (1994) [39]: I think the year will be deleted and the correct is Maggi and colleagues [39]. Please modify their similarities through the manuscript

Corrected as suggested.

Reviewer 2 Report

Comments for Authors:

This manuscript presents information of interest to experts related to the subject; however, I consider that it is important and necessary to improve its wording based on the following general and specific comments.

General comments:

1. The Abstract is extensive, I suggest leaving only what is important or relevant to the research and sticking to the maximum word format (200).

2. This study could have been complemented with the quantification of mycotoxins, of which it is asserted that their production is inhibited derived from the mitigation of hyphal formation.

3. The application at the industrial level could have been reinforced if data on the proliferation of this fungus in the food matrix during its shelf life after the removal of the sample from the hyperbaric storage treatment were included.

Specific comments:

Title

The title is confusing, it is suggested to re-write it for better understanding. Some suggestions could be:

Byssochlamys nivea ascospores: Development control and inactivation by hyperbaric storage at room temperature”

“Control and inactivation of the development of Byssochlamys nivea ascospores by hyperbaric storage at room temperature”

Abstract

Line 23, the word "noteworthy", sounds a bit pretentious. Please consider removing or replacing.

Lines 27-37, from line 27 the ideas become confusing and repetitive. It is suggested to be mostly concise and precise to improve the writing.

Keywords

Line 38, to increase the scope and search rank of the manuscript, it is suggested that the keywords are not included in the title of the manuscript.

Introduction

Although it is clearly structured and complete, the Introduction is extensive and not very concise, it is suggested to work on it. For example, the first two paragraphs (lines 42-56) can be reduced to just one, based essentially on the Ascomycota clade and the risk that its spores imply in contaminating food given its resistant characteristics.

Something similar to what was previously mentioned happens in lines 57-70, although the paragraph has a clear idea and is adequately structured, it offers very extensive information, so it is recommended to reduce it to the spore germination process and the production of toxins through from vegetative structures.

The following paragraphs of the Introduction (lines 71-109) contain useful information; however, it is suggested to review the relevance of this information in the Introduction, since there are some details that could be more useful in the discussion of the manuscript.

Results

Line 233, …”it was even observed a smaller increase” … This idea is confusing, please clarify it!

Line 254, …“an apparent recovery was observed from the 5th day onwards for both the total microbial load and total ascospore load”… Please clarify this idea, it is confusing!

It is suggested to review the relevance of the paragraph in lines 371-375, it seems a "premature conclusion", which could be more useful in other sections of the document such as in the discussion or conclusion.

Lines 382-385, since the journal separates the discussion results sections, it is important that this section limit itself exclusively to showing the results. The inferences and deductions that it raises are very interesting, but they would be more useful and impactful in the results section.

The Figures mentioned on lines 471-472 (Figures 3C, 6C and 8C) do not appear in the document. Please include or delete this sentence!

Conclusions

The conclusions should be more concise and focus them on the objectives of the investigation.

Author Response

  1. The Abstract is extensive, I suggest leaving only what is important or relevant to the research and sticking to the maximum word format (200).

Changed accordingly. The abstract section was revised to meet the maximum word limit. Please check the abstract section with the highlighted changed left as red-colour text.

  1. This study could have been complemented with the quantification of mycotoxins, of which it is asserted that their production is inhibited derived from the mitigation of hyphal formation.

The authors agree with the referee. The quantification of mycotoxins could have complemented the present work. However, the present study aimed to provide a first insight on how HS/RT could be used to control the development of B. nivea ascospores. The formation of mycotoxins will be accessed in the future, as the experimental work is still ongoing. Yet, phase contrast microscopy analysis can provide a quick indication on hyphae formation, and, indirectly, mycotoxin formation. Additionally, as the ascospores do not develop (under HS conditions), the risk of mycotoxin formation is eliminated, considering that hyphae formation would be necessary for the production of mycotoxins.

  1. The application at the industrial level could have been reinforced if data on the proliferation of this fungus in the food matrix during its shelf life after the removal of the sample from the hyperbaric storage treatment were included.

The authors agree with the referee. A large-scale study should be performed in the future to understand the industrial relevance of the findings of the present manuscript. Yet, this would require a very intense use of the laboratorial scale high pressure equipment (with limited vessel capacity), which would require several months to obtain all the samples and would also imply that other works that are currently being performed in  different fields to be fully stopped to make room for such amount of samples. Yet, we will leave a statement on the manuscript, namely on the future perspectives, stating the validation of the present findings at an industrial scale level.

Specific comments:

Title

The title is confusing, it is suggested to re-write it for better understanding. Some suggestions could be: “Byssochlamys nivea ascospores: Development control and inactivation by hyperbaric storage at room temperature” or “Control and inactivation of the development of Byssochlamys nivea ascospores by hyperbaric storage at room temperature”.

Changed accordingly. The title was changed to: “Development control and inactivation of Byssochlamys nivea ascospores by hyperbaric storage at room temperature”. The authors acknowledge the referee for the kind suggestions.

Abstract

Line 23, the word "noteworthy", sounds a bit pretentious. Please consider removing or replacing.

Changed accordingly.

Lines 27-37, from line 27 the ideas become confusing and repetitive. It is suggested to be mostly concise and precise to improve the writing.

Changed accordingly. The whole abstract was rewritten to meet the journals’ standards. Please check the abstract section.

Keywords

Line 38, to increase the scope and search rank of the manuscript, it is suggested that the keywords are not included in the title of the manuscript.

Changed accordingly.

Introduction

Although it is clearly structured and complete, the Introduction is extensive and not very concise, it is suggested to work on it. For example, the first two paragraphs (lines 42-56) can be reduced to just one, based essentially on the Ascomycota clade and the risk that its spores imply in contaminating food given its resistant characteristics.

Changed accordingly.

Something similar to what was previously mentioned happens in lines 57-70, although the paragraph has a clear idea and is adequately structured, it offers very extensive information, so it is recommended to reduce it to the spore germination process and the production of toxins through from vegetative structures.

Changed accordingly. The information in the Lines 57-70 was shortened as suggested.

The following paragraphs of the Introduction (lines 71-109) contain useful information; however, it is suggested to review the relevance of this information in the Introduction, since there are some details that could be more useful in the discussion of the manuscript.

The authors generally agree with the referee. Some of the informations provided in Lines 71-109 regarded experimental results on the application of intense thermal processing, and its combination with HPP to enhance ascospore inactivation. Thus, these informations were moved from the original location at the introduction section and moved to the discussion section. Please check the manuscript with the changes made highlighted as red-colour text.

Results

Line 233, …”it was even observed a smaller increase” … This idea is confusing, please clarify it!

Changed accordingly. Now it reads, in lines 231-233: “…storage experiments, despite of being only observed a slight increase (p<0.05) in the total microbial load (of about 0.32 log units), while for samples kept under refrigeration, for the same period, it was observed an even smaller increase (p<0.05) of 0.19 log units…”.

Line 254, …“an apparent recovery was observed from the 5th day onwards for both the total microbial load and total ascospore load”… Please clarify this idea, it is confusing!

Changed accordingly. Now it reads, in lines 253-258: “…nevertheless, an increase on the total microbial load and total ascospore load was observed (of about 0.5 and 0.6 log units, respectively) from the 5th day onwards. This in-crease in total ascospore load is most likely due to the release of entrapped ascospores that could have remained inside the asci as a consequence of the harvesting process [5,29], as it is also accompanied by the increase of the thermal resistant fraction (Figure 1B).”.

It is suggested to review the relevance of the paragraph in lines 371-375, it seems a "premature conclusion", which could be more useful in other sections of the document such as in the discussion or conclusion.

Changed accordingly. The information previously placed in lines 371-375 was removed from the results’ section and placed in the discussion section (now available in Lines 568-570).

Lines 382-385, since the journal separates the discussion results sections, it is important that this section limit itself exclusively to showing the results. The inferences and deductions that it raises are very interesting, but they would be more useful and impactful in the results section.

Changed accordingly. The information available in Lines 382-385 were removed from the results’ section and placed in the discussion section (Lines 556-559). Now it reads, in Lines 556-559: “Regarding HPP pre-activated samples, it was observed a reduction on the thermal resistant fraction right after the HPP process, which can be related to structural damages on the crust, according to the literature, which will ultimately facilitate the ascospore hydration and an efficient pressure-transmission afterwards [6]. In addition, it is…”.

The Figures mentioned on lines 471-472 (Figures 3C, 6C and 8C) do not appear in the document. Please include or delete this sentence!

The Figures are placed in the manuscript, but we mistakenly changed the number of the figures. We intended to mention Figures 2C, 5C and 7C, which regard the phase contrast microscopy images. The manuscript was corrected.

Conclusions

The conclusions should be more concise and focus them on the objectives of the investigation.

Changed accordingly. The conclusion section was revised to be more concise and focused on the topic.
